# Pediatric Malaria with Respiratory Distress: Prognostic Significance of Point-of-Care Lactate

**DOI:** 10.3390/microorganisms11040923

**Published:** 2023-04-02

**Authors:** Catherine Mitran, Robert O. Opoka, Andrea L. Conroy, Sophie Namasopo, Kevin C. Kain, Michael T. Hawkes

**Affiliations:** 1Faculty of Medicine, University of Alberta, Edmonton, AB T6G 2R3, Canada; mitran@ualberta.ca; 2Department of Paediatrics and Child Health, Mulago Hospital and Makerere University, Kampala P.O. Box 7062, Uganda; 3Ryan White Center for Pediatric Infectious Diseases and Global Health, Indiana University School of Medicine, Indianapolis, IN 46202, USA; 4Department of Paediatrics, Kabale District Hospital, Kabale P.O. Box 1102, Uganda; 5Sandra Rotman Centre for Global Health, Department of Medicine, University Health Network-Toronto General Hospital, University of Toronto, Toronto, ON M5G 1L7, Canada; 6Department of Laboratory Medicine and Pathobiology, University of Toronto, Toronto, ON M5G 1L7, Canada; 7Department of Paediatrics, Faculty of Medicine, University of Alberta, Edmonton, AB T6G 2R3, Canada; 8Department of Medical Microbiology and Immunology, Faculty of Medicine, University of Alberta, Edmonton, AB T6G 2R3, Canada; 9School of Public Health, University of Alberta, Edmonton, AB T6G 1C9, Canada; 10Stollery Science Lab, Edmonton, AB T6G 1C9, Canada; 11Women and Children’s Health Research Institute, Edmonton, AB T6G 1C9, Canada

**Keywords:** malaria, *Plasmodium falciparum*, lactate, mortality, respiratory distress, hazard ratio, Africa, child

## Abstract

Respiratory distress (RD) in pediatric malaria portends a grave prognosis. Lactic acidosis is a biomarker of severe disease. We investigated whether lactate, measured at admission using a handheld device among children hospitalized with malaria and RD, was predictive of subsequent mortality. We performed a pooled analysis of Ugandan children under five years of age hospitalized with malaria and RD from three past studies. In total, 1324 children with malaria and RD (median age 1.4 years, 46% female) from 21 health facilities were included. Median lactate level at admission was 4.6 mmol/L (IQR 2.6–8.5) and 586 patients (44%) had hyperlactatemia (lactate > 5 mmol/L). The mortality was 84/1324 (6.3%). In a mixed-effects Cox proportional hazard model adjusting for age, sex, clinical severity score (fixed effects), study, and site (random effects), hyperlactatemia was associated with a 3-fold increased hazard of death (aHR 3.0, 95%CI 1.8–5.3, *p* < 0.0001). Delayed capillary refill time (τ = 0.14, *p* < 0.0001), hypotension (τ = −0.10, *p* = 0.00049), anemia (τ = −0.25, *p* < 0.0001), low tissue oxygen delivery (τ = −0.19, *p* < 0.0001), high parasite density (τ = 0.10, *p* < 0.0001), and acute kidney injury (*p* = 0.00047) were associated with higher lactate levels. In children with malaria and RD, bedside lactate may be a useful triage tool, predictive of mortality.

## 1. Introduction

In 2021, malaria killed an estimated 619,000 people worldwide [1]. Most of these deaths were caused by *Plasmodium falciparum* infection in children under the age of five in sub-Saharan Africa. Of the human malaria-causing species, *P. falciparum* is the most virulent, owing largely to its extensive antigenic variation and ability to sequester in the microvasculature of host tissues [2,3]. Individuals living in high transmission settings typically develop clinical immunity to severe falciparum malaria by adolescence [4]. However, prior to acquiring immunity, children are at significant risk of severe malaria, resulting in a disproportionate burden of morbidity and mortality in the pediatric population.

Severe falciparum malaria is a complex disease that affects multiple organ systems, with symptoms varying widely depending on multiple host and parasite factors. Clinical presentations of severe malaria include cerebral malaria (CM), severe malaria anemia (SMA), and malaria with respiratory distress (RD) [5]. In children, RD is associated with the highest mortality rate of the three syndromes, although many patients present with overlapping symptoms [5]. In a study of 1844 Kenyan children with malaria, the mortality in isolated RD was 24% compared to 7.3% in isolated CM and 1.3% in SMA, respectively [5]. Unlike in adults, where respiratory distress is generally a result of inflammatory-mediated pulmonary edema, in children, respiratory distress is largely attributed to underlying metabolic acidosis [6,7,8,9]. Metabolic acidosis in severe malaria results from the build-up of ketone bodies and organic acids, the most significant of which is lactic acid [10,11,12,13]. 

Hyperlactatemia at admission is an important predictor of mortality in pediatric malaria patients [5,6,7,14,15,16,17]. Critically ill malaria patients in resource-limited settings are often managed without access to a central laboratory. Clinical signs such as deep (acidotic) breathing may be used as indirect clinical correlates of lactic acidosis [18]. However, the severity of these signs can fluctuate over short periods of time, with substantial interobserver variability [19]. Point-of-care (POC) devices offer a convenient method of lactate measurement at the bedside that does not rely on a central laboratory or on subjective judgement. Previous studies have demonstrated the prognostic utility of admission lactate levels in pediatric malaria patients in resource-limited settings [16,20,21]. However, these studies did not focus on RD, which is an important subgroup of severe malaria, associated with high mortality. Furthermore, these studies were performed in large referral hospitals, limiting the generalizability of these findings to peripheral health facilities. Validation of POC lactate testing in rural and remote health facilities is important, as access to central laboratory assays (e.g., blood gases) is limited. 

Lactate accumulation may result from increased lactate production or impaired clearance. In malaria, hyperlactatemia has been attributed to lactate production by hypoxic host tissues and *Plasmodium* parasites, as well as impaired renal and hepatic clearance of lactate. While hypotension leading to systemic tissue hypoperfusion is thought to be a major contributor to hyperlactatemia in sepsis [22], hypotension is less common in malaria patients. Thus, global tissue hypoperfusion has not been previously associated with hyperlactatemia in malaria [18]. However, systemic tissue hypoxia and hyperlactatemia can result from reduced oxygen carrying capacity in SMA [23]. A unique feature of malaria is the sequestration of infected erythrocytes in host microvasculature and rosetting of parasitized erythrocytes with uninfected RBCs. This can obstruct blood flow causing local areas of tissue hypoxia and anaerobic glycolysis, resulting in increased lactate production [24,25,26]. The etiology of hyperlactatemia in malaria is further complicated by the fact that *Plasmodium* parasites produce lactate as a metabolic by-product [27]. This source of lactate may be important in patients with hyperparasitemia [28]. Acute kidney injury (AKI) and liver dysfunction may occur as complications of severe malaria [29,30,31,32]. Under physiological conditions, the liver removes approximately 70% of lactate and the kidneys remove the remaining 30% from circulation through gluconeogenesis and direct oxidation [33,34]. End organ damage in severe malaria may therefore lead to decreased lactate clearance and hyperlactatemia. 

The objective of this study was to examine whether POC measurement of blood lactate can provide a simple, accurate means of risk-stratifying pediatric patients with malaria and RD in a variety of clinical settings (referral hospital and peripheral health facilities). As a secondary objective, we explored associations between the lactate level and other clinical and laboratory parameters to develop hypotheses about the relative contribution of different potential mechanisms of lactate buildup in malaria with RD.

## 2. Materials and Methods

### 2.1. Study Design and Patient Inclusion Criteria

We carried out a secondary analysis of children hospitalized with malaria and RD, combining data from three different studies in Uganda. Study 1 was a randomized controlled trial of inhaled nitric oxide as adjunctive treatment for severe malaria (July 2011 to June 2013, *n* = 180) [35]. In this secondary analysis, we included only patients from the placebo group, who were treated according to standard of care without experimental adjunctive therapy. Study 2 was a prospective observational study of children with febrile illness presenting to a single regional referral hospital (February 2012 to August 2013, *n* = 2502) [36]. Study 3 was a prospective study of children with hypoxemia presenting to 20 rural hospitals or health centres (June 2019 to November 2021, *n* = 2405) [37]. Patients were included in the secondary analysis if they: (1) were under five years of age; (2) were hospitalized; (3) had fever at admission; (4) tested positive for *P. falciparum* by malaria rapid diagnostic test (mRDT) or by microscopy; (5) had objective signs of respiratory distress; and (6) had a POC blood lactate measurement recorded at admission.

### 2.2. Setting

Uganda is a high transmission setting for *P. falciparum* and is estimated to account for 5% of malaria cases globally [38]. There is heterogenous spatial distribution of malaria incidence across the country, with mean monthly incidence rates of *P. falciparum* infection in different regions ranging from 7.9 to 52.3 cases per 1000 per month [39]. There is seasonal variation in incidence, with the peak occurring between the months of June and July and the lowest trough seen in February and March [39].

### 2.3. Sample Size

A sample size calculation indicated that we would need at least 740 patients for our primary analysis. To estimate this sample size, we used package gsDesign [40] in the R statistical environment [41] for a time-to-event analysis with 2 arms (hyperlactatemic patients versus controls) with 80% power at the α = 0.05 level of significance. We assumed: mortality among controls of 8% by 14 days [35]; hazard ratio of death associated with hyperlactatemia of 2 [42]; ratio of hyperlactatemic patients to controls of 1:2 [35]; and censoring rate of 100% by day 14 (i.e., all patients discharged, transferred, or died by day 14). 

### 2.4. Study Procedures

Participants were diagnosed with malaria using mRDTs targeting the *P. falciparum* antigen histidine-rich protein 2 (HRP2) and the pan-*Plasmodium* antigen parasite lactate dehydrogenase (pLDH), according to manufacturer’s guidelines. A test was considered positive if either the HRP2 or the pLDH band were positive. Microscopy was not performed for all patients included in this study. However, among patients with a microscopy result available, semiquantitative parasitemia estimates were determined by analysis of thick smears, prepared using Field’s stain. For Study 1, parasite density was quantified using the ratio of trophozoites to leukocytes on the thick smear, assuming 8000 leukocytes/µL. For Studies 2 and 3, parasite density was graded by the clinical laboratory at each site, according to the “plus” scale: + (1 to 9 trophozoites in 100 fields); ++ (1 to 10 trophozoites in 10 fields); +++ (1 to 10 trophozoites per field); ++++ (>10 trophozoites per field). The grade of parasitemia was used to estimate the order of magnitude of parasite density: +  =  50 parasites/µL; ++  =  500 parasites/µL, +++  =  5000 parasites/µL; ++++ = 50,000 parasites/µL [43]. Patients who tested positive for malaria were treated according to national guidelines. POC lactate levels were measured from a finger-prick blood sample using the Lactate Scout analyzer (Sports Resource Group, Inc., Minneapolis, MN, USA) [44]. For Study 1, creatinine was measured from venous blood at the bedside using the i-STAT CHEM8+ cartridge on the i-STAT handheld biochemistry analyzer (Abbott Point of Care, Mississauga, ON, Canada). For Study 2, creatinine was quantified using the modified Jaffe colorimetric method on an Alinity c instrument (Abbott, Lake Forest, IL, USA).

### 2.5. Clinical Definitions

Respiratory distress was defined as the presence of any of the following signs: nasal flaring, intercostal or subcostal retractions, or abnormally deep (acidotic) breathing [5]. Hyperlactatemia was defined as peripheral blood lactate measurement >5 mmol/L [18]. Fever was defined as presenting with a history of fever in the past 48 h (reported by the caregiver) or axillary temperature > 37.5 °C. Hypoxemia was defined as oxygen saturation (SpO_2_) less than 90% as measured using a fingertip pulse oximeter. 

Hypotension was defined as a systolic blood pressure (*SBP*) below the lower limit of the normal range for age (5th percentile at 50th height percentile), using the clinical formula [45]: $$SBP[mmHg]=70+(2\times age\left[yr\right])$$

To estimate global oxygen delivery and blood oxygen content, we used the sum of the hemoglobin-bound and dissolved oxygen, according to the following equations [46]. The stroke volume was estimated based on findings from a previous study of Ghanaian children with malaria [47]: $$D_{O_{2}}=C_{O_{2}}\times Qt$$
$$C_{O_{2}}=\left(Hb\times Sp_{O_{2}}\times 1.34\right)+(P_{O_{2}}\times 0.23)$$
Qt=SV×HR
SV=0.0408×BSA
where



$D_{O_{2}}:\mathrm{oxygen}\mathrm{delivery}\left[\mathrm{mL}/\min\right]$





$Qt:\mathrm{cardiac}\mathrm{output}\left[L/\min\right]$





$C_{O_{2}}:\mathrm{blood}\mathrm{oxygen}\mathrm{content}\left[\mathrm{mL}/L\right]$





$Hb:\mathrm{hemoglobin}\mathrm{concentration}\left[g/L\right]$





$Sp_{O_{2}}:\mathrm{oxygen}\mathrm{saturation}$





$P_{O_{2}}:\mathrm{partial}\mathrm{pressure}\mathrm{of}\mathrm{dissolved}\mathrm{oxygen}\;[\mathrm{kPa}]$





SV:strokevolume [L]





$HR:\mathrm{heart}\mathrm{rate}\;[\min^{-1}]$





$BSA:\mathrm{body}\mathrm{surface}\mathrm{area}[m^{2}]$



Body surface area was estimated using the formula [48]:$$BSA\;[m^{2}]=\sqrt{\frac{wt\;[kg]\times ht\;[cm]}{3600}}$$

In the absence of blood gas measurement in our resource-limited setting, we imputed the partial pressure of dissolved oxygen from the oxygen saturation [49]. Severe malarial anemia (SMA) was defined as hemoglobin <50 g/L [18]. Acute kidney injury (AKI) was defined according to the Kidney Disease: Improving Global Outcomes (KDIGO) guidelines [50]. Children were diagnosed with AKI if there was an increase in creatinine of ≥1.5 × baseline or a eGFR < 35 mL/min/1.73 m^2^, which was assumed to have occurred over the past 7 days [50]. A height-independent approach was used to estimate the baseline creatinine, assuming a GFR of 120 mL/min/1.73 m^2^, as previously described [51,52]. A clinical risk score was calculated for each participant using the Lambaréné Organ Dysfunction Score (LODS) [53]. The LODS was calculated based on three variables: prostration, coma, and deep breathing, with each variable assigned a score of one if present, resulting in possible scores ranging from one to three [53].

### 2.6. Statistical Analysis

Descriptive data were summarized using the median and interquartile range (IQR) for continuous variables or the number (%) for binary variables. Following dichotomization based on lactate level at admission, Kaplan–Meier curves were used to compare survival between the normal and high lactate populations. To determine if hyperlactatemia was an independent predictor of mortality, we used a linear mixed-effects Cox proportional hazard model, with adjustment for age, sex, and clinical severity (LODS) as fixed effects. Because the lactate was assessed across 3 studies and 21 sites, we adjusted for study and site as random effects in the model. We used package *coxme* [54] in the R statistical environment [41] for the survival analysis. The area under receiver operator characteristic (AUROC) curves were used to compare the prognostic value of lactate, LODS, SpO_2_, and RR at admission. The lactate cut-off that most accurately distinguished fatal from nonfatal cases was determined from ROC curve analysis (Youden index). Odds ratios and the corresponding 95% confidence intervals were calculated in the R statistical environment [41]. Nonparametric methods (Mann–Whitney *U* test) were used to examine associations between variables for continuous data and categorical data were analyzed using Pearson’s two-tailed chi-square or Fisher’s exact test, as appropriate. For nonparametric rank correlation coefficients between variables, we used Kendall’s tau (τ) as a robust method that did not require normal distribution of data and was permissive for tied ranks [55]. Analysis and visualization of data were performed using GraphPad Prism version 6 (GraphPad Software Inc., USA), and the R statistical environment (version 4.2.2) [41].

### 2.7. Ethical Approval and Informed Consent

Written informed consent was obtained from the parent or guardian of all study participants prior to enrolment in the primary study. The primary studies were approved by Makerere University School of Medicine Research Ethics Board, Kampala, Uganda (REC Protocol 2010-107 and 2011-255), the Makerere University School of Biomedical Sciences Research Ethics Committee, Kampala, Uganda (REC Protocol SBS 644), the University Health Network Research Ethics Board, Toronto, Canada (REB Number 10-0607-B and 12-0039-AE), and the University of Alberta Health Research Ethics Board (Pro00084784).

## 3. Results

We included 1324 children with malaria and RD (median age 1.4 years, 46% female) hospitalized across 21 centres in Uganda from February 2012 to November 2021. We included 49 patients from Study 1, 603 patients from Study 2, and 680 patients from Study 3 that met the inclusion criteria (Appendix A). Clinical characteristics of the pooled cohort are shown in Table 1. 

### 3.1. Handheld Blood Lactate Predicts Subsequent Mortality in Children with Malaria and RD

The median lactate level at admission was 4.6 mmol/L (IQR 2.6–8.5) and 586 patients (44%) had hyperlactatemia (lactate > 5 mmol/L). Mortality in Studies 1, 2, and 3 was 12.0%, 9.0%, and 3.7%, respectively, and the pooled mortality was 84/1324 (6.3%). There were 67 deaths (11%) among children with hyperlactatemia at admission and 17 deaths (2.3%) among children with normal lactate (*p* < 0.0001). Hyperlactatemia was associated with a crude HR of mortality of 5.3 (95%CI 3.1–9.0, *p* < 0.0001, Figure 1A). After adjustment for age, sex, clinical severity score (fixed effects), study, and site (random effects), the adjusted HR (aHR) was 3.0 (95%CI 1.8–5.3, *p* < 0.0001, Table 2). In the subgroup of 690 patients with parasitemia confirmed by microscopy, the aHR was 4.0 (95%CI 1.7–9.2, *p* < 0.0001). Subgroup analysis by study is shown in Figure 1B. 

We next used ROC curve analysis to compare lactate with other predictors of mortality (Figure 2). The AUROC for lactate was 0.74 (95%CI 0.68–0.80, *p* < 0.0001). The optimal cut-off (threshold of maximum discrimination) using the Youden index for the ROC curve was 5.7 mmol/L (sensitivity 62%, specificity 79%). Lactate had similar discriminatory power to the LODS, a composite clinical severity score (AUROC 0.74 vs. 0.75, *p* = 0.80, Figure 2A,B). In combination with LODS, lactate improved the prediction of mortality over LODS alone (AUROC 0.80 vs. 0.75, *p* = 0.010). With respect to other clinical signs, SpO_2_ was not a statistically significant predictor of subsequent mortality (AUROC 0.53, *p* = 0.44, Figure 2C). Respiratory rate (RR) was a weaker predictor of mortality (AUROC 0.60, *p* = 0.0038, Figure 2D). 

### 3.2. Lactate Clearance as a Predictor of Mortality

Among 586 patients with hyperlactatemia at admission, a subgroup of 164 (28%) had a second lactate measurement obtained at 8–12 h after admission, allowing for the early lactate clearance to be computed. Of note, a second lactate measurement was available in only 5 (7.4%) fatal cases, compared to 159 (31%) nonfatal cases (*p* < 0.0001). Eight patients (4.9%) had a rising lactate level and 90 (55%) had slow lactate clearance, based on a first-order decay half-life greater than 10.8 h [16,42]. There were 2 deaths among children with delayed lactate clearance and 3 deaths among children with normal lactate clearance (*p* = 0.39). 

### 3.3. Signs of Increased Work of Breathing Are Associated with Hyperlactatemia and Mortality

Clinical signs of respiratory distress associated with hyperlactatemia included nasal flaring (OR 2.4, 95%CI 1.7–3.5), subcostal retractions (OR 2.0, 95%CI 1.4–2.8), and deep (acidotic) respirations (OR 1.9, 95%CI 1.5–2.3) (Figure 3A). The same signs were also associated with an increased chance of mortality: subcostal retractions (OR 3.3, 95%CI 1.6–7.3), nasal flaring (OR 2.9, 95%CI 1.4–6.9), and deep (acidotic) respirations (OR 2.1, 95%CI 1.3–3.8) (Figure 3B). Tachypnea was statistically associated with hyperlactatemia but not mortality (Figure 3). Grunting was associated with mortality but not hyperlactatemia (Figure 3). Wheezing was statistically associated with lower odds of hyperlactatemia and was not associated with mortality. 

### 3.4. Elevated Lactate Is Associated with Impaired Tissue Perfusion

Delayed capillary refill time is a clinical sign of impaired tissue perfusion. Lactate levels rose monotonically with increasing capillary refill time (τ = 0.14, *p* < 0.0001, Figure 4A). In a subset of 568 patients in whom systolic blood pressure was recorded, systolic hypotension (decompensated shock) was present in 14 (2.4%). The lactate was inversely correlated with the systolic blood pressure (τ = −0.10, *p* = 0.00049). Lactate was higher among patients with hypotension (9.4 mmol/L, IQR 4.9–16) compared to patients without hypotension (4.8 mmol/L, IQR 2.4–11, *p* = 0.039).

### 3.5. Elevated Lactate Is Not Associated with Hypoxemia

In patients with RD, impaired lung function and gas exchange may result in hypoxemia and impaired tissue oxygenation. Of 1315 patients with available data for SpO_2_, 667 patients (51%) were hypoxemic (SpO2 < 90%). We did not observe a statistically significant correlation between lactate levels and peripheral oxygen saturation (τ = 0.016, *p* = 0.40, Figure 4B). 

### 3.6. Elevated Lactate Is Associated with Anemia and Low Global Oxygen Delivery

Hemoglobin is the major carrier protein for the bulk transfer of oxygen from the lungs to the tissues. In a subset of 810 patients in whom a hemoglobin measurement was performed, 448 (55%) were severely anemic (hemoglobin < 50 g/L). The lactate level was inversely correlated with the hemoglobin (τ = −0.25, *p* < 0.0001, Figure 4C). Patients with SMA had 3.4-fold higher odds (95%CI 2.5–4.6) of hyperlactatemia compared to patients with hemoglobin ≥ 50 g/L. Similarly, the lactate was higher among patients who required a blood transfusion (6.7 mmol/L (IQR 3.8–12) versus 3.4 mmol/L (IQR 2.2–5.5), *p* < 0.0001).

We examined tissue oxygenation using the $D_{O_{2}}$, an index that accounts for hemoglobin-bound oxygen, dissolved oxygen, and cardiac output. We found that $D_{O_{2}}$ was inversely correlated with lactate level (τ = −0.19, *p* < 0.0001, Figure 4D). 

### 3.7. Elevated Lactate Is Associated with Higher Parasite Density

*P. falciparum* parasites may generate L- and D-lactate, contributing to the total circulating lactate levels [27]. Furthermore, increased sequestered parasite biomass in capillary beds may contribute to tissue hypoxia via impaired perfusion. In a subset of 855 patients in whom the circulating parasite density was quantified by light microscopy, lactate increased with increasing semiquantitative parasite count (τ = 0.10, *p* < 0.0001, Figure 4E). Using previously published quantitative estimates of parasite and host lactate production [28], we calculated that parasite-derived lactate accounted for a median of 0.28% (IQR 0.028 to 2.8) of the total lactate production. Parasites accounted for 10% or more of the total lactate production in 58/848 (6.8%) patients (Appendix A).

### 3.8. Association with Acute Kidney Injury

Lactate is mainly cleared through metabolic pathways in both the liver and kidney [28]. We did not have data on hepatic function; however, 224 patients had creatinine measurements, allowing us to assess kidney function. In this subgroup, there were 114 patients (51%) with AKI. The lactate level was statistically significantly correlated with the creatinine level (τ = 0.17, *p* < 0.0001). Patients with AKI had higher levels of lactate (median 6.6 mmol/L, IQR 2.4 to 12) than patients with normal kidney function (median 3.6 mmol/L, IQR 2.0 to 7.0), *p* = 0.00047, Figure 4F).

## 4. Discussion

RD is associated with high mortality in patients with malaria [5] and risk-stratification tools could improve clinical management. Here, we demonstrate that POC lactate measurement at admission predicts subsequent mortality in pediatric patients with malaria and RD and that implementation of bedside lactate measurement is feasible across a variety of clinical settings in an LMIC. POC lactate measurement consistently risk-stratified patients in a randomized controlled trial, an observational study at a large referral hospital, and at 20 rural health facilities across Uganda. This study is noteworthy for its focus on malaria with RD, which is less frequently studied than other complications of severe malaria, such as CM and SMA. Furthermore, the large number of patients in this study (*n* = 1324) and the multiple health facilities from which they were recruited increases the generalizability of our findings. 

### 4.1. Clinical Significance of Hyperlactatemia in Malaria with RD

Patients with hyperlactatemia at admission had a significantly higher hazard of death compared to those with normal lactate levels. In a multivariable analysis adjusting for age, sex, disease severity, and site, the hazard of death was threefold higher in patients with hyperlactatemia at admission. The estimate of the HR was similar between the 3 primary studies included in this secondary analysis (Figure 1B), despite differences in acuity between the referral centre (Studies 1 and 2, mortality 12% and 9.0%) and rural health facilities (Study 3, mortality 3.7%). This suggests that lactate is a clinically informative prognostic tool in malaria with RD across multiple settings. These findings are consistent with previous studies that have demonstrated the prognostic utility of blood lactate measurement in other severe malaria syndromes [5,6,7,14,15,16,17]. Our study includes a larger number of patients than past studies and was the first to focus on malaria with RD. Our study, like most others on severe malaria, used a cut-off for hyperlactatemia of >5 mmol/L, in accordance with the WHO definition of severe malaria [18]. Using ROC curve analysis on our dataset, we determined that the optimal cut-off (Youden index) was lactate >5.7 mmol/L, which supports the use of the WHO threshold. This value is similar to the optimal cut-off determined by Aramburo et al. (>5.2 mmol/L) for children with malaria in Eastern Africa [16]. 

Respiratory distress in pediatric malaria is associated with metabolic acidosis and hyperlactatemia [18]. We found that deep breathing, nasal flaring, and subcostal retractions were associated with higher odds of hyperlactatemia. These clinical signs were also associated with increased odds of mortality, highlighting the central role of lactate in the progression of malaria with RD. The only respiratory sign that was associated with lower odds of having hyperlactatemia was wheezing, which was not associated with mortality. In these patients, *P. falciparum* parasitemia may be an incidental finding and not the primary etiology of respiratory distress, as wheezing is more commonly a symptom of asthma or viral bronchiolitis. Our study informs clinical practice by providing evidence for signs that healthcare workers can use in low-resource settings to identify patients at increased risk of hyperlactatemia and death.

Lactate clearance rate has been proposed as a prognostic indicator of mortality in pediatric malaria patients [16]. Although a POC handheld device, such as the Lactate Scout analyzer, would be ideal for serial bedside monitoring of lactate, we found that these measurements had limited clinical utility outside of the referral hospital. In our study, patients at highest risk of mortality either required transfer to a larger centre or died before a second lactate measurement could be taken (8–12 h after admission). Thus, we were unable to examine the association of lactate clearance with mortality, due to the small number of fatalities in which serial lactate measurements were available. A previous study on sequential lactate measurement in severe malaria reported that 49.6% of deaths occurred within the first 8 h of admission, before a second lactate measurement could be taken [16]. However, in patients who present with high lactate levels and survive past 8 h, serial lactate measurement may provide insight into response to treatments and prognosticate risk of later mortality. Sustained high lactate concentrations in children at 8 h was strongly associated with all-cause mortality at 72 h [16]. It is also possible that early, frequent measurements of lactate could be informative; however, further studies are required to determine if these measurements would offer prognostic value beyond a single lactate measurement at admission. 

There are several potential use cases for POC lactate in clinical settings in LMICs. First, lactate may be useful as a stand-alone triage tool. In our study, lactate was as accurate as LODS for the prediction of mortality (Figure 2). Clinical severity scores, such as LODS, based on clinical signs alone, can be calculated in resource-limited settings; however, they may be susceptible to observer error. There are inherent challenges with assessing clinical signs of respiratory distress in children, which can result in large interobserver variability [19,58]. Thus, a convenient and objective tool, such as POC lactate, may be desirable in settings where health workers have limited pediatric assessment skills. Second, in the hands of skilled pediatricians, POC lactate could improve triage accuracy beyond what can be achieved by clinical assessment alone. After adjustment for clinical severity score, hyperlactatemia remained a statistically significant independent predictor of mortality (Table 2). Furthermore, in our ROC curve analysis, when POC lactate measurement was combined with LODS, there was a statistically significant improvement in mortality prediction. These findings are similar to a previous study, which reported that addition of laboratory values, including lactate, improved the predictive value of a clinical score in pediatric malaria patients [59]. Further study is required to determine if this could impact patient outcomes. By highlighting the utility of POC lactate as a prognostic tool, our study may serve as a model to expand the use of bedside laboratory tools to improve the management of childhood malaria.

Key advantages of POC lactate include objectivity of the measurements, simplicity of use, and minimal training requirements. The Lactate Scout analyzer is simple to operate and uses a finger prick blood sample to measure lactate levels, making it accessible in community settings. The device is designed to be used by nonexperts, as it is marketed to athletes to monitor blood lactate levels during training. Thus, nurses at rural health facilities in our study were able to collect accurate lactate measurements with minimal training. POC lactate measurements obtained by multiple users across different settings were equally predictive of mortality (Figure 1B), demonstrating the feasibility and value of integrating this device with current patient management protocols. The relatively low cost of the test (USD 2.42 [42]) suggests that large-scale implementation may be financially viable. These measurements could help nurses and community healthcare workers identify children at the highest risk of death to ensure that they receive prompt referral and treatment. Additionally, POC lactate measurement could augment healthcare workers’ clinical decision making, leading to more effective healthcare resource allocation. We speculate that this may result in an overall cost saving. Evaluation of cost-effectiveness will be an important part of future studies using this device. 

### 4.2. Mechanisms of Hyperlactatemia in Malaria with RD

As a secondary objective, we explored associations between the lactate level and other clinical and laboratory parameters (Figure 4). Based on these associations, we may develop hypotheses about different mechanisms of lactate accumulation in the circulation among children with malaria and RD. Several pathophysiological processes may contribute to hyperlactatemia in malaria with RD, such as lactate production by host tissues or by *Plasmodium* parasites, and impaired lactate clearance [28]. We found that higher blood lactate levels were associated with delayed capillary refill time, hypotension, lower hemoglobin concentration, lower oxygen delivery, higher parasite burden, and AKI, but not with oxygen saturation.

Anaerobic cellular respiration resulting in lactate as a metabolic by-product may occur when tissue oxygen demands exceed the circulatory supply. In severe systemic inflammatory diseases, distributive shock may lead to tissue hypoxia through impaired systemic perfusion. In our study, we used capillary refill time as a clinical surrogate of global tissue perfusion. We found a statistically significant association between increased lactate levels and prolonged capillary refill time. Consistent with this, median lactate levels were significantly higher in the small number of patients (2.4%) with overt systolic hypotension (decompensated shock). Delayed capillary refill may also be observed in hypovolemic shock due to dehydration. Children with hyperlactatemia were more likely to present with symptoms suggestive of dehydration including vomiting and being unable to drink or breastfeed. Although distributive and hypovolemic shock contribute to hyperlactatemia in patients with bacterial sepsis, their role in severe malaria is controversial (reviewed in [60]). Accordingly, liberal fluid resuscitation in severe malaria has little effect on lactic acidosis and may be contraindicated [18,61]. Cardiac ultrasonography in children with severe malaria generally demonstrates increased stroke volume and cardiac output, reflecting high output status [47].

In addition to impaired perfusion, inadequate tissue oxygenation may be related to low oxygen-carrying capacity due to anemia. In our study of children with malaria and RD, comorbid SMA was common (55%). Lactate levels were inversely correlated with hemoglobin concentrations and those with SMA had 3.4-fold higher odds of having hyperlactatemia. These findings are consistent with those reported in the literature; however, there is evidence that the association between hemoglobin and lactate levels differs, depending on the clinical syndrome. While most studies have found an inverse relationship between hemoglobin and lactate in malaria [15,62], a study by Brand et al. reported an association in children with SMA, but not in children with cerebral malaria [23]. 

In malaria with RD, tissue hypoxia and lactate production may occur when blood oxygen content falls due to impaired gas exchange at the level of the lungs (hypoxemia). It is common for adult malaria patients to develop noncardiogenic pulmonary edema, leading to impaired lung function [8]. In our study, 50% of patients with RD were hypoxemic. However, we did not find a significant correlation between lactate concentration and peripheral oxygen saturation. Instead, anemia was more closely associated with impaired oxygen transport and hyperlactatemia. We speculate that, in children with RD, impaired oxygen exchange at the alveolar-capillary unit is a minor contributor to hyperlactatemia, relative to low oxygen carrying capacity (anemia), and impaired tissue perfusion.

Global oxygen delivery ($D_{O_{2}}$) is an index of oxygen transport to the tissues that integrates several of the above factors, including hemoglobin-bound oxygen, dissolved oxygen, and the cardiac output ($Q_{t}$) [46]. In our study, oxygen delivery was inversely correlated with lactate level, suggesting that tissue hypoxia contributes significantly to hyperlactatemia. The effect of corrective actions (administration of supplemental oxygen and intravenous fluids) cannot be assessed in our study but is an area of active research [61].

Lactate production may be attributable to specific end organs, such as the central nervous system. Altered mental status and lethargy were significantly higher in children with hyperlactemia (Table 1). Brain tissue ischemia in CM may result in lactate production, which may occur at higher rates than in other tissues [63]. Neurologically derived lactate may thus be an important source of circulating lactate.

Besides systemic factors, local tissue hypoxia, caused by sequestration of parasitized erythrocytes, may contribute to increased lactate production in severe malaria [24,25,26]. Lactate levels were significantly higher in patients with a higher parasite burden, suggesting that sequestered parasites may be occluding capillaries, leading to local tissue hypoxia and higher lactate production. However, total and sequestered parasite burdens are not strongly correlated with the circulating parasitemia [64]. Proteins from the PfEMP1 family, which mediate sequestration to host tissues, are only found on the surface of erythrocytes infected with mature malaria parasites. As a result, the peripheral parasitemia may over- or underestimate the sequestered parasite biomass, depending on the synchrony of the parasite lifecycles, the average age of the parasite population at the time of examination, and the degree of host immunity [64]. Some studies have employed techniques to directly observe parasite sequestration, such as orthogonal polarization spectral imaging of rectal mucosa [24] or sidestream dark-field imaging of sublingual microcirculation [26]. It would be of interest to use these techniques to examine the association of parasite sequestration with circulating lactate levels. 

*Plasmodium* parasites themselves may contribute to the pool of circulating lactate [28]. Using previously published quantitative estimates of lactate production rates by infected erythrocytes and by host tissues, we calculated that parasites contributed only a small fraction (median 0.28%) of the total lactate production (Appendix A). However, lactate production from parasites can be substantial under conditions of high parasite density [28], as was observed in a minority (6.9%) of patients in our study, in whom parasite lactate production exceeded 10% of the total. Our estimate differs from a previous computation, in which parasite lactate production was 655 mmol/day (7.4%) of the total lactate production of 8800 mmol/day in children with severe malaria [28]. Our calculation used more realistic estimates of body weight and blood volume in children, whereas the previous calculation used adult body weight and blood volume [28]. We also accounted for variability in the parasite biomass, using the observed parasitemia and the estimated ratio of sequestered to circulating parasites, whereas the previous study assumed that 10% of body erythrocytes were parasitized [28]. On the other hand, the semiquantitative parasitemia used in our study, based on the “plus” system [43], may have underestimated the circulating parasite density. Thus, the circulating parasitemia was median 500 iRBC/µL in our study compared to 40,000 iRBC/µL in the AQUAMAT multicentre study [65].

Decreased hepatic and renal clearance of lactate may represent another mechanism of hyperlactatemia. We did not collect data on liver function, but we were able to assess kidney function in a subset of 224 patients. Lactate clearance by the kidney occurs via metabolic conversion of lactate to pyruvate (gluconeogenesis) and urine excretion. Approximately 30% of the lactate is removed via gluconeogenesis in the proximal tubules of the renal cortex under physiologic conditions [33]. Urinary excretion of lactate is significant only when lactate levels exceed 6 mol/L [28]. AKI is common in both adult and pediatric patients with severe malaria [29,30,31]. AKI in severe malaria is a multifactorial disorder caused by intravascular hemolysis, microvascular obstruction, immune-mediated injury, endothelial cell dysfunction, and hemodynamic instability [31,66]. In the subset of patients in whom creatinine measurements were available, 51% had AKI. Patients with AKI had significantly higher blood concentrations of lactate at admission, compared to those with normal kidney function. This is consistent with a previous study of severe pediatric malaria in which higher blood lactate levels were associated with increasing severity of AKI [29]. 

Additional mechanisms of hyperlactatemia, not assessed in our study, may warrant further research. For example, malaria is associated with reduced gut barrier integrity, as evidenced by increased levels of intestinal injury markers in children with malaria [67,68]. A recent study demonstrated an association between elevated lactate and gut injury marker intestinal fatty acid binding protein (I-FABP) [68]. Gut ischemia due to parasite sequestration in the splanchnic circulation may explain increased host-derived lactate. Alternatively, L- and D-lactate of gut microbial origin may contribute to the circulating lactate pool. Furthermore, organic acids other than lactate, some of bacterial origin, have prognostic significance in malaria [69]. Unmeasured acids may contribute to metabolic acidosis and mortality in malaria and may explain why the lactate is not perfectly predictive of subsequent death in our study. 

Through systematic analysis of clinical and laboratory factors associated with hyperlactatemia, our study adds to the current understanding of the underlying mechanisms that lead to elevated lactate levels. 

### 4.3. Study Limitations

This was a secondary analysis of individual patient data from three previous studies in Ugandan children. These studies were not designed to address our specific research question; nonetheless, similarities between the patients enrolled in the three studies allowed us to assemble a large, pooled cohort of patients with well-defined malaria and RD, all of whom had measurements for lactate at admission and prospective follow-up for subsequent mortality. We addressed potential heterogeneity in the three studies with a subgroup analysis (Figure 1B) and adjustment for study as a random effect in our primary analysis. Our study focused on children with malaria and RD; therefore, our results should not be extrapolated to other malaria syndromes or to severe malaria in general. Lactate measurements were obtained using the POC device without parallel confirmatory assays using conventional laboratory protocols. However, we have previously demonstrated the accuracy of this device in a field setting [70] and described its prognostic value in pediatric pneumonia in a similar setting [42]. The definition of malaria used for inclusion of patients in this analysis was detection of parasite antigen (mRDT) or detection of parasites on microscopy. This inclusive definition could have resulted in false positive classification of patients with malaria. To address this, we performed a subgroup analysis of 690 patients with microscopy-confirmed parasitemia and showed that the prognostic value of lactate remained significant in this subgroup (aHR 4.0). The median oxygen saturation among patients in our study was 89%, which is lower than other studies of severe childhood malaria. This may relate to RD as an inclusion criterion in our study. Inclusion of patients with a wider range of oxygen saturations may have allowed us to detect an association between oxygen saturation and lactate level. For several secondary analyses, some data were incomplete because they were not collected in the primary studies (e.g., hepatic function, which could have provided additional insight into lactate clearance in our patients). Some investigations, such as arterial blood gases and orthogonal polarization spectral imaging, were not available in our resource-limited setting. Thus, in many cases we relied on more subjective measures, such as capillary refill time as a surrogate of tissue perfusion. Some physiologic parameters, such as the stroke volume and partial pressure of dissolved oxygen, were not measured directly but were estimated based on formulae from past studies. Of note, these estimates would only affect our secondary analysis of the relationship between lactate and oxygen delivery (Figure 4D). Clinical observations were performed by multiple healthcare workers at different sites, which may have affected the precision of these variables [19]. On the other hand, the prognostic value of admission lactate was established across settings and users, supporting the robustness and generalizability of our findings. Additionally, the retrospective and observational design of the study precluded us from determining if POC lactate measurement could improve patient outcomes. A randomized controlled study of lactate-guided therapy for malaria with RD would be of interest to define the impact on patient outcomes. 

## 5. Conclusions

Here, we demonstrated that POC lactate measurement at admission predicts mortality in pediatric patients with malaria and RD across a range of studies (RCT, single-centre, and multicentre prospective cohort studies) and settings (pediatric referral hospital and rural health facilities). POC lactate testing offers a simple and objective means by which to risk-stratify patients, particularly in locations without access to a central laboratory. Furthermore, in the 20 rural health facilities, nurses performed POC lactate measurements with little oversight from study staff, demonstrating the feasibility of implementing POC blood lactate testing in a wide range of low-resource settings. The addition of this measurement to current triage algorithms in resource-limited settings could improve the efficiency of healthcare resource allocation by identifying children at highest risk of mortality. In areas such as sub-Saharan Africa, where malaria is a leading cause of death in children under five years of age, combining POC lactate measurements with existing triage protocols may offer a strategy to reduce childhood mortality. 

## Figures and Tables

**Figure 1 microorganisms-11-00923-f001:**
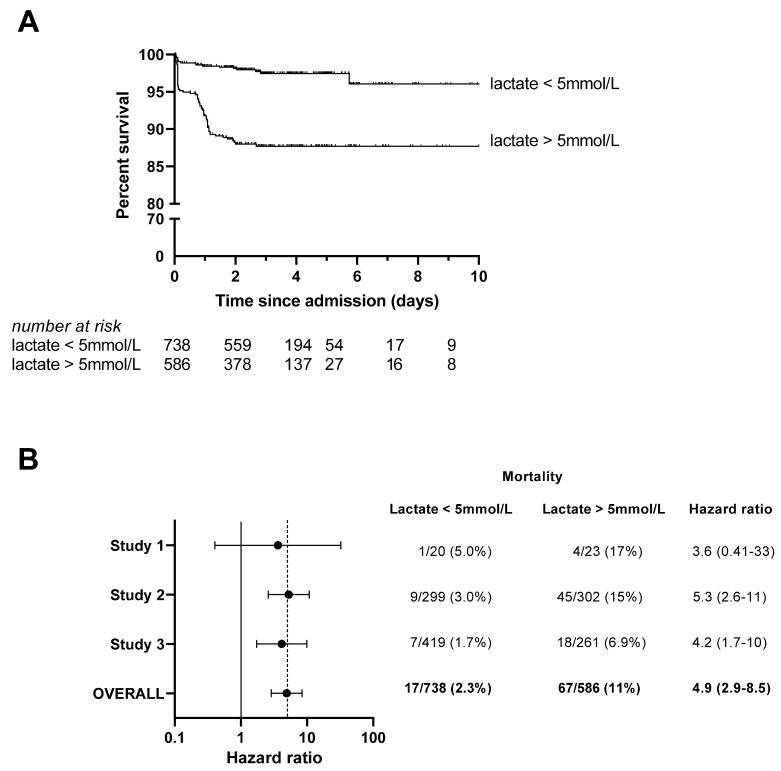
Point-of-care lactate measurement at admission predicts mortality in pediatric malaria patients with respiratory distress. (**A**) Kaplan–Meier curves display the survival of children stratified by presence or absence of hyperlactatemia (lactate > 5 mmol/L) at admission. Ticks denote patients that were right-censored as they were discharged from hospital, absconded, or transferred to another facility. Values at the bottom of the figure denote the number of participants remaining in each group at different timepoints. The crude hazard ratio was 5.3 (95%CI 3.1–9.0, *p* < 0.0001). (**B**) Hazard ratios for each primary study contributing to the pooled analysis were similar, with confidence intervals reflecting the sample size of each study. The pooled hazard ratio (random effects model) was 4.9 (95%CI 2.9–8.5, *p* < 0.0001).

**Figure 2 microorganisms-11-00923-f002:**
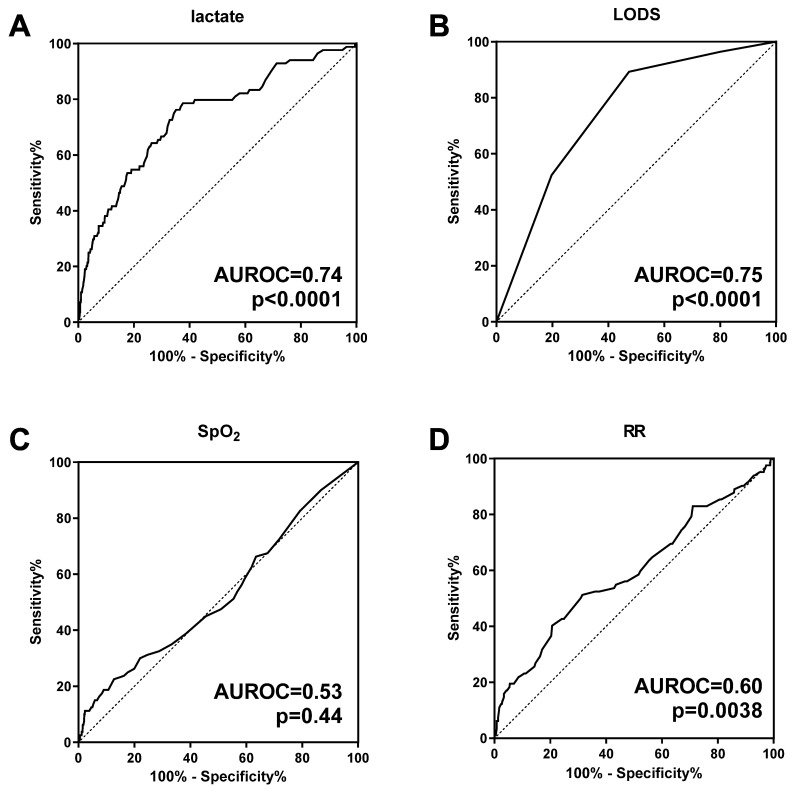
Lactate measurements at admission discriminate fatal from nonfatal malaria with respiratory distress. Receiver operator characteristic (ROC) curves were generated for (**A**) blood lactate concentration at admission, (**B**) Lambaréné Organ Dysfunction Score (LODS), (**C**) oxygen saturation (SpO_2_), and (**D**) respiratory rate (RR). Area under ROC (AUROC) and *p*-values are provided for each curve. There was no statistically significant difference between the AUROC for blood lactate versus LODS (AUROC 0.74 vs. 0.75, *p* = 0.80).

**Figure 3 microorganisms-11-00923-f003:**
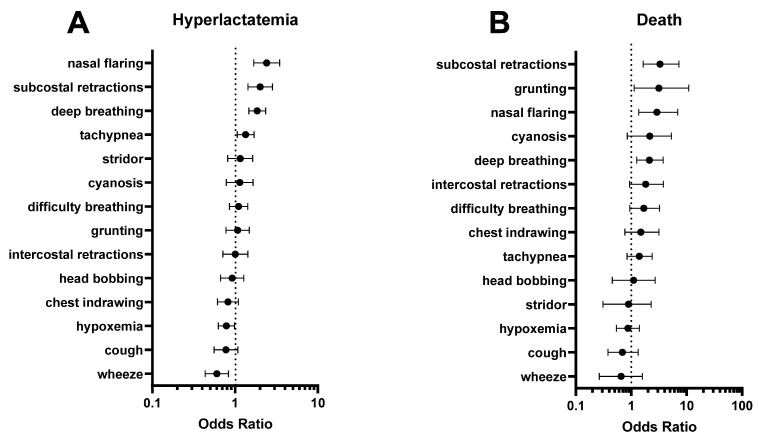
Association of clinical signs of respiratory distress with (**A**) hyperlactatemia and (**B**) death. Of note, deep (acidotic) respirations, nasal flaring, and subcostal retractions were associated with both hyperlactatemia and mortality. Closed circles represent the calculated odds ratio, and bars represent the 95% confidence interval.

**Figure 4 microorganisms-11-00923-f004:**
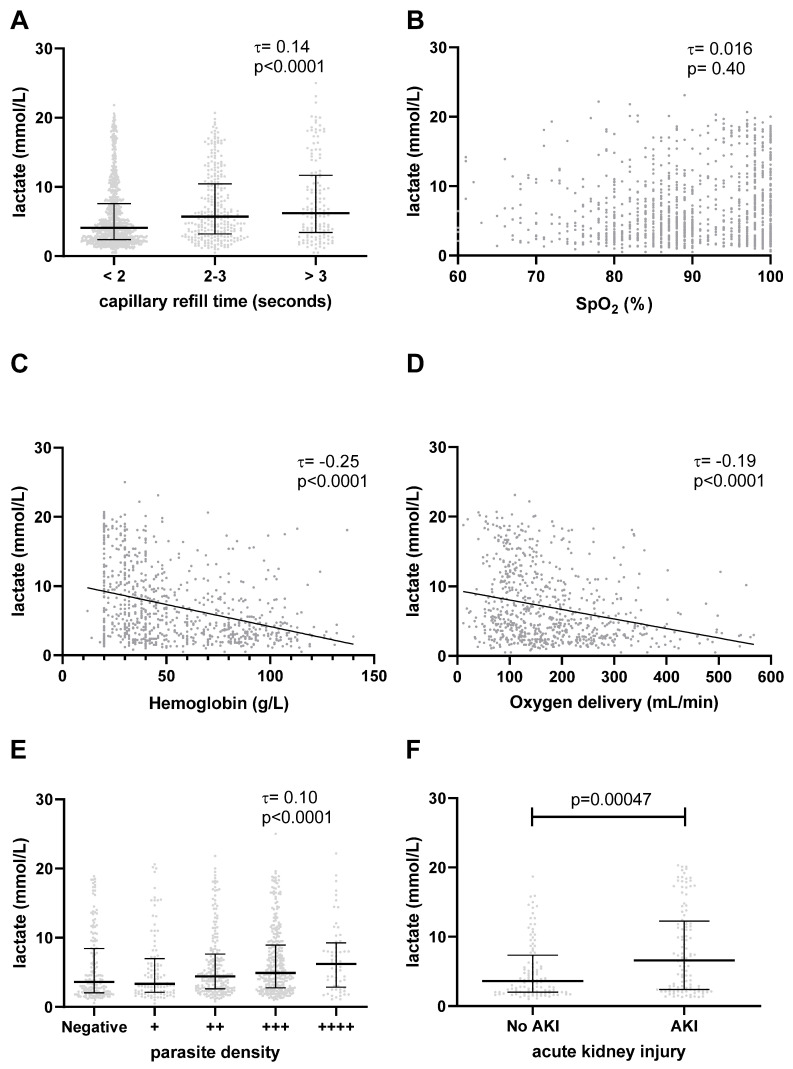
Mechanistic insights derived from the association of lactate with other clinical and laboratory variables. (**A**) Patients with prolonged capillary refill time had higher blood lactate concentrations than patients with normal capillary refill times. (**B**) Lactate was not correlated with oxygen saturation (SpO_2_). (**C**) Hemoglobin concentration and (**D**) global oxygen delivery ($D_{O_{2}}$) were inversely correlated with lactate levels. (**E**) Higher circulating parasite densities were associated with higher lactate levels. (**F**) patients with AKI had significantly higher blood lactate concentrations than patients without AKI. Together, these data suggest that impaired tissue oxygenation (poor perfusion, low blood oxygen carrying capacity, and local hypoxia due to parasite sequestration) as well as impaired renal lactate clearance contribute to hyperlactatemia. Kendall’s tau (τ) was used to determine nonparametric rank correlation coefficients between variables.

**Table 1 microorganisms-11-00923-t001:** Characteristics at hospital admission of 1324 Ugandan children with malaria and respiratory distress, stratified by bedside blood lactate level.

	Entire Cohort (*n* = 1324)	Lactate ≤5 mmol/L (*n* = 738)	Lactate >5 mmol/L (*n* = 586)	*p*-Value
** *Demographics* **				
Female sex	611 (46)	354 (48)	257 (44)	0.14
Age [years], median (IQR)	1.4 (0.75–2.4)	1.2 (0.75–2.3)	1.4 (0.75–2.5)	0.43
** *History* **				
Cough	1147 (87)	649 (88)	498 (85)	0.14
Difficulty breathing	975 (74)	537 (73)	438 (75)	0.48
Lethargy	438 (33)	195 (26)	243 (42)	<0.0001
Convulsions	329 (25)	180 (24)	149 (25)	0.71
Unable to feed/drink	687 (52)	342 (46)	345 (59)	<0.0001
Vomiting	524 (40)	257 (35)	267 (46)	<0.0001
Diarrhea	352 (27)	188 (26)	164 (28)	0.35
** *Physical examination findings* **				
Weight [kg], median (IQR)	9 (7.7–12)	9 (7.5–12)	9.5 (8–12)	0.38
Wasting ^1^	202 (16)	114 (16)	88 (16)	>0.99
Length/height [cm], median (IQR)	72 (65–83)	72 (65–83)	72 (65–84)	0.87
Stunting ^2^	651 (51)	358 (50)	293 (53)	0.37
Heart Rate (bpm), median (IQR)	160 (140–170)	150 (130–170)	160 (140–180)	0.00036
Tachycardia ^2^	767 (58)	390 (53)	377 (65)	<0.0001
Respiratory rate (bpm), median (IQR)	52 (42–62)	50 (40–60)	54 (44–62)	0.0015
Tachypnea ^3^	834 (64)	444 (61)	390 (68)	0.013
Oxygen saturation (%), median (IQR)	89 (84–98)	89 (84–98)	90 (85–98)	0.098
Hypoxemia (SaO_2_ < 90%)	667 (51)	392 (53)	275 (47)	0.033
Temperature [°C], median (IQR)	38 (37–38)	38 (37–39)	38 (37–38)	<0.0001
Fever ^4^	798 (61)	470 (64)	328 (56)	0.0045
Altered level of consciousness ^5^	458 (35)	197 (27)	261 (45)	<0.0001
Capillary refill time [seconds]				<0.0001
<2	888 (68)	545 (75)	343 (59)	
2–3	286 (22)	128 (18)	158 (27)	
>3	141 (11)	58 (7.9)	83 (14)	
Chest indrawing	1086 (82)	615 (83)	471 (80)	0.19
Wheeze	368 (54)	247 (59)	121 (46)	0.0018
Stridor	206 (30)	122 (29)	84 (32)	0.45
Danger signs ^6^	1070 (81)	567 (77)	503 (86)	<0.0001
** *Treatment* **				
Artesunate	771 (58)	440 (60)	331 (56)	0.27
Quinine	434 (33)	226 (31)	208 (35)	0.069
Artemether injection	41 (3.1)	14 (1.9)	27 (4.6)	0.0076
Artemether-lumefantrine (oral)	143 (11)	88 (12)	55 (9.4)	0.16
Supplemental oxygen	681 (53)	399 (56)	282 (50)	0.058
** *Outcome* **				<0.0001
Discharged	1043 (79)	627 (86)	416 (71)	
Death	84 (6.4)	17 (2.3)	67 (11)	
Transferred to another facility	101 (7.7)	46 (6.3)	55 (9.4)	
Absconded	89 (6.8)	43 (5.9)	46 (7.9)	

Data represent *n* (%) unless otherwise specified; IQR, interquartile range; ^1^ weight-for-height/length below −3SD [56]; ^2^ height/length-for-age below -3SD [56]; ^3^ vital sign >99th percentile for age [57]; ^4^ axillary temperature >37.5 °C; ^5^ less than “alert” on AVPU scale; ^6^ 1 or more of the following: vomiting, convulsions, unable to feed/drink, altered consciousness, severe malnutrition (weight-for-age z-score less than −3), or stridor at rest.

**Table 2 microorganisms-11-00923-t002:** Univariable and multivariable hazard ratios for mortality among children with malaria and respiratory distress.

	Univariable		Multivariable *	
	HR	*p*-Value	aHR	*p*-Value
Fixed effects				
Female sex	0.68 (0.43–1.1)	0.093	0.77 (0.48–1.2)	0.25
Age	0.99 (0.82–1.2)	0.38	1.0 (0.85–1.3)	0.73
LODS		<0.0001		<0.0001
0	1.0 (reference)		1.0 (reference)	
1	1.2 (0.29–4.8)		1.6 (0.36–6.7)	
2	7.0 (2.1–23)		9.0 (2.5–33)	
3	14 (4.5–47)		18 (5.0–65)	
Lactate > 5 mmol/L	5.3 (3.1–9.0)	<0.0001	3.0 (1.8–5.3)	<0.0001
Random effect				
Site			Variance 1.2	

* Cox mixed-effects proportional hazard model, adjusting for age, sex, and LODS as fixed effects and for site as random effect.

## Data Availability

Data are available upon request from the study authors.

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
