# Peer review of "Pediatric Malaria with Respiratory Distress: Prognostic Significance of Point-of-Care Lactate"

_microorganisms, 2023, doi:10.3390/microorganisms11040923_

Round 1

Reviewer 1 Report

1. The secondary objective (associations between the lactate level and other clinical and laboratory parameters to develop hypotheses of lactate buildup in malaria with RD) is not well discussed. Authors are suggested to modify the discussion accordingly.

2. In the discussion authors repeatedly mentioned that many of the findings were in line with previous studies. This approach questions the novelty of this study. Authors are suggested to focus on their results, not on the limitations. They should modify the discussion to emphasize the significance of their own findings.  

Reviewer 2 Report

Pediatric malaria with respiratory distress: prognostic significance of point-of-care lactate

The authors present work correlating lactic acidosis with outcomes in children with respiratory distress in the setting of severe malaria in a variety of settings.  The secondary objective is to identify potential pathophysiologic contributors to hyperlactemia in severe malaria patients. The manuscript is well written and includes a very large patient population. Furthermore, contributors to hyperlactemia are well addressed and add significantly to the literature.

Major Points

Abstract

The authors state “Lactic acidosis contributes to the pathophysiology of malaria with RD”’.  Please explain further.  Lactic acidosis is a biomarker of severe disease but how do the authors propose it contributes to the pathology of RD in the disease.

Introduction

The authors state “In children, RD is associated with the highest mortality rate of the three syndromes, although many patients present with overlapping symptoms.”  Please provide a reference in support of this as well as actual mortality rates in the cited manuscripts.  While it is true that ARDS in adult patients with malaria portends an exceptionally poor prognosis, with mortality rates quoted up to 90%, pediatric literature is much less clear on the mortality of isolated RD outside of SMA/CM.  

Methods

The initial part of the methods section does not talk about slide positivity for inclusion in the study and instead states that MRDT positivity was the requirement.  Further down in methods, good detail is given regarding the microscopy for each of the three studies.  Was microscopy considered in the diagnosis?  IE if the patient was MRDT positive but microscopy negative were they still included?  This is particularly relevant for group 3, where patients presented with hypoxia.  In these patients in the absence of positive microscopy it would be much more likely that their disease was pneumonia or bronchiolitis.  This would then lead to lactic acidosis for other reasons than are seen in malaria (i.e. more likely dehydration, hypoxia itself, shock). 

For calculation of blood oxygen content and oxygen delivery, measurement of SV is necessary.  No mention in the methods discusses how SV was measured in the included patients.

Results

Cohort mean oxygen saturation is 89% which is significantly lower than the vast majority of patients presented with severe malaria.  Perhaps the reason hypoxia was not associated with lactate levels was that across the cohort, oxygenation was poor (rather than a lack of association if a wider spread of saturations was seen).  Consider including this in the limitations.

Given the potential role of lactate clearance contributing to hyperlactemia in patients with AKI, I would like to see Cr analyzed as a continuous variable rather than an absolute categorical variable (yes vs no AKI).  I

The results state that the risk factors “Unable to feed/drink and vomiting” are significantly worse in those with hyperlactemia.  If you have measured a SV to calculate oxygen delivery, how does SV and CI compare to lactate?  I.E. do you have sonographic evidence of hypovolemia.

Discussion

Altered mental status and convulsions are also significantly higher in children with hyperlactemia. These children may be the subset of patients that have co-existing CM.  During brain injury/brain tissue ischemia, significant amounts of lactate are produced (at higher rates than in other tissues).  Consider adding to the discussion the potential role of neurologically derived lactate.

Its unclear to me how much hyperlactemia improves prognosis in a cohort of children with RD beyond the presence of RD itself.  As the cohort included only those with RD, this is impossible to determine at this point and it should be noted as a limitation

Round 2

Reviewer 1 Report

Authors are suggested to modify the conclusion according to the changes made in the discussion.

Reviewer 2 Report

Issues addressed per my request.  Congratulations on a nice article.